# Reanalysis of cryo-EM data reveals ALK-cytokine assemblies with both 2:1 and 2:2 stoichiometries

**Jan Felix** [1,2]*, **Steven De Munck**[1,2], **J. Fernando Bazan**[3], **Savvas N. Savvides**[1,2]*

**1** VIB-UGent Center for Inflammation Research, Ghent, Belgium, **2** Unit for Structural Biology, Department of Biochemistry and Microbiology, Ghent University, Ghent, Belgium, **3** ℏ bioconsulting llc, Stillwater, Minnesota, United States of America

* savvas.savvides@ugent.be (SNS); jan.felix@ugent.be (JF)

## Abstract

Activation of Anaplastic lymphoma kinase (ALK) and leukocyte tyrosine kinase (LTK) by their cognate cytokines ALKAL2 and ALKAL1 plays important roles in development, metabolism, and cancer. Recent structural studies revealed ALK/LTK-cytokine assemblies with distinct stoichiometries. Structures of ALK-ALKAL2 and LTK-ALKAL1 complexes with 2:1 stoichiometry determined by X-ray crystallography contrasted the 2:2 ALK-ALKAL2 complexes determined by cryo-EM and X-ray crystallography. Here, we show based on reanalysis of the cryo-EM data deposited in EMPIAR-10930 that over half of the ALK-ALKAL2 particles in the dataset are classified into 2D and 3D classes obeying a 2:1 stoichiometry besides the originally reported structure displaying 2:2 stoichiometry. Unlike particles representing the 2:2 ALK-ALKAL2 complex, particles for the 2:1 ALK-ALKAL2 complex suffer severely from preferred orientations that resulted in cryo-EM maps displaying strong anisotropy. Here, we show that extensive particle orientation rebalancing in cryoSPARC followed by 3D refinement with Blush regularization in RELION constitutes an effective strategy for avoiding map artifacts relating to preferred particle orientations and report a 3D reconstruction of the 2:1 ALK-ALKAL2 complex to 3.2 Å resolution from EMPIAR-10930. This new cryo-EM structure together with the crystal structures of ALK-ALKAL2 and LTK-ALKAL1 complexes with 2:1 stoichiometry reconciles a common receptor dimerization mode for ALK and LTK and provides direct evidence for the presence of an ALK-ALKAL2 complex with 2:1 stoichiometry next to the reported 2:2 stoichiometric assembly in the EMPIAR-10930 dataset. Finally, our analysis emphasizes the importance of public deposition of raw cryo-EM data to allow reanalysis and interpretation.

**Data availability statement:** The cryo-EM map and accompanying structural model for the 2:1 ALKTG-EGFL-ALKAL2 complex have been deposited in the EMDB/PDB with accession codes: EMD-51087/9g5i. The particle .star file obtained after 3D refinement with Blush Regularization in RELION 5 is added to this manuscript as a Supplementary Data file (S1 File) and can be used to extract particles after running CTF estimation on the Motion Corrected and Dose Weighted (*_DW.mrc) micrographs available from EMPIAR-10930 (https://www.ebi.ac.uk/empiar/EMPIAR-10930/).

**Funding:** SNS acknowledges research support from the Flanders Institute for Biotechnology ("Vlaams Instituut voor Biotechnologie", VIB, grant number C0101) and the FWO ("Fonds voor Wetenschappelijk Onderzoek – Vlaanderen", grant number G0B4918N). The funders had no role in study design, data collection and analysis, decision to publish, or preparation of the manuscript.

**Abbreviations:** ADP, atomic displacement parameter;ALK, Anaplastic lymphoma kinase; CSF-1R, colony stimulating factor 1 receptor; EGFR, epidermal growth factor receptor; FGFR, fibroblast growth factor receptor; Flt3R, Flt3 receptor; FSC, Fourier Shell Correlation; IR, insulin receptor; LTK, leukocyte tyrosine kinase; PDGFR, platelet-derived growth factor receptor; RTK, receptor tyrosine kinase; VEGFR, vascular endothelial growth factor receptor.

## Introduction and results

Anaplastic lymphoma kinase (ALK) and the related leukocyte tyrosine kinase (LTK) are receptor tyrosine kinases (RTKs) that are activated by binding their cognate cytokines, ALKAL1 and ALKAL2 (also called FAM150A/B or AUGβ/α) [1–3]. Their signaling outputs elicit a range of pleiotropic activities in development and metabolism, and their dysregulated intracellular kinases are keenly pursued as therapeutic drug targets in various cancers [4]. However, the field had long been hampered by a paucity of insights into the intricate folds of ALK and LTK and their cytokine-bound complexes. A triad of recent studies [5–7] have aimed to provide the missing structural and mechanistic framework for the cytokine-mediated activation of ALK family receptors but diverged in their proposed assemblies.

Our study by De Munck *and colleagues* [5] provided representative assemblies for the entire ALK family by reporting crystal structures of both the ligand-binding extracellular regions of human ALK and LTK (ALK$_{TG}$/LTK$_{TG}$) in complex with ALKAL2 and ALKAL1, respectively (PDB entries 7nwz and 7nx0). These structures revealed how a single cytokine molecule nucleates homodimeric receptor assembly via distinct site 1 and 2 interaction interfaces, creating a key site 3 receptor-receptor contact (S1 Fig). In accordance with the evolutionary relatedness [8] and close structural similarity between ALK and LTK, key receptor-receptor and receptor-ligand interactions are conserved in the resulting 2:1 stoichiometric ALK-ALKAL2 and LTK-ALKAL1 assemblies [5], which are pseudo-symmetric due to the asymmetric nature of the ALKAL2/1 ligands. These structures thus suggest an asymmetric mode of ALK/LTK receptor dimerization and activation akin to fibroblast growth factor receptor (FGFR) signaling [9], and contrasting with symmetric receptor dimerization as observed for the RTKs Kit [10,11], colony stimulating factor 1 receptor (CSF-1R) [12], Flt3 receptor (Flt3R) [13], epidermal growth factor receptor (EGFR) [14], platelet-derived growth factor receptor (PDGFR) [15], and vascular endothelial growth factor receptor (VEGFR) [16].

In contrast, Li *and colleagues* [6] and Reshetnyak *and colleagues* [7] have focused only on ALK-ALKAL2 complexes comprising the extracellular ligand-binding fragment of ALK and the membrane-proximal EGFL domains (ALK$_{TG-EGFL}$), and reported symmetric assemblies obeying a 2:2 stoichiometry obtained via X-ray crystallography using an ALK$_{TG-EGFL}$-ALKAL2 fusion construct (PDB entry 7LS0) or separate ALK$_{TG-EGFL}$ and full-length ALKAL2 via electron cryo-microscopy (cryo-EM, PDB entry 7n00), respectively (S1B Fig). In this side-by-side architecture, the main cytokine-receptor interface corresponds to the high affinity site 1 also reported by De Munck *and colleagues* [5]. An additional ALK-ALKAL2' interaction, covering 250 Å$^2$ of buried surface area, was proposed to be responsible for bridging the two 1:1 ALK-ALKAL2 subcomplexes into an assembly that could support signaling. Intrigued by this 2:2 stoichiometry differing from the 2:1 stoichiometry revealed by our crystal structures of truncated ALK$_{TG}$ in complex with ALKAL2 (and LTK$_{TG}$ with ALKAL1), we closely examined the cryo-EM data and analysis reported by Reshetnyak *and colleagues* [7] as deposited in the EMPIAR [17] data base (entry EMPIAR-10930).

At the outset, we identified a number of 3D-classes (classes 1, 3, 7, 8, and 9 illustrated in Extended Data Fig 4 in Reshetnyak *and colleagues* [7]) resembling the compact 2:1 $ALK_{TG}/LTK_{TG}$-cytokine complexes we had reported [5]. To investigate this apparent particle heterogeneity, we leveraged the raw cryo-EM data and uncleaned particle stack as deposited by Reshetnyak *and colleagues* [7] (access code EMPIAR-10930). An initial data processing strategy following the methods described in Reshetnyak *and colleagues* [7], including 8× binning of the data (Materials and methods) revealed that 6 of the 10 most populated classes after initial 2D classification do not correspond to classes shown in the final 2D classification reported in Reshetnyak *and colleagues* [7] (Fig 1A). This prompted us to pursue extensive data processing (Materials and methods, S2 Fig) of particles present in both sets of 2D classes, namely the ones that correspond to clear 2-fold symmetry (Fig 1A, orange squares) and the ones that lack apparent symmetry (Fig 1A, blue squares). However, the final refined map of the C1 symmetric particles displayed severe preferential particle orientations leading to a map that is smeared along the Z-direction, a common problem in cryo-EM requiring sample and data collection optimization. Although the resulting map clearly displays two $ALK_{TG-EGFL}$ extracellular domains and one ALKAL2 copy in a similar 2:1 assembly as observed in the crystal structure of the $ALK_{TG}$-ALKAL2 complex (S2 Fig), artifacts resulting from severe anisotropy hampered confident model building and prohibited detailed structural interpretations.

To overcome the apparent anisotropy presented in the 3D reconstructions toward a 2:1 $ALK_{TG-EGFL}$–ALKAL2 complex, we reasoned that combining exhaustive automatic particle orientation rebalancing (available in cryoSPARC [18], v4.5) to exclude particles from over-populated direction bins, followed by 3D refinement with Blush regularization [19] in RELION v5, might result in more isotropic 3D reconstructions. Blush regularization uses a pre-trained denoising convolutional neural network on pairs of half-set reconstructions as a smoothness prior and is shown to remove artifactual densities resulting from uneven angular distributions or streaky features in overfitted maps [19]. This new regularization approach to resolving vexing cryo-EM structural data was successfully employed to reveal the distinct binding modes of the anti-CRISPR repressor Aca2 with DNA and RNA [21].

Starting from a set of selected 3D classes (1.4 million particles) after 3D hetero-refinement (Materials and methods), we divided these 3D classes in two sets based on the absence or presence of apparent 2-fold symmetry. Such grouping resulted in 3D classes resembling either the C1-symmetric 2:1 $ALK_{TG}$-ALKAL2 crystal structure reported by De Munck *and colleagues* [5] (876,806 particles, ~60% of the total), or the deposited C2-symmetric 2:2 $ALK_{TG-EGFL}$-ALKAL2 cryo-EM map reported by Reshetnyak *and colleagues* [7] (523,467 particles, ~40% of the total) (Fig 2A).

Firstly, we wondered whether our approach would lead to a cryo-EM map of similar resolution and information content for the C2 symmetric $ALK_{TG-EGFL}$-ALKAL2 complex with 2:2 stoichiometry as reported by Reshetnyak *and colleagues* [7]. Indeed, further data processing in cryoSPARC resulted in a 2.3 Å resolution map for the 2:2 $ALK_{TG-EGFL}$-ALKAL2 complex (Materials and methods). The resulting map has a comparable resolution as the deposited map by Reshetnyak *and colleagues* [7] (EMD-24095, 2.3 Å) and, importantly, does not contain any particles from 2D classes corresponding to a 2:1 $ALK_{TG-EGFL}$-ALKAL2 stoichiometry (Fig 2A). We note that, similarly as the cryoSPARC refined map reported in Reshetnyak *and colleagues* [7], map signal for the EGFL domains is less defined than for the RELION refined map obtained by Reshetnyak *and colleagues* [7] at slightly lower resolution.

We then pursued analysis of the particle set with apparent resemblance to $ALK_{TG-EGFL}$-ALKAL2 complexes with 2:1 stoichiometry (Materials and methods). The combined use of Orientation Rebalancing in CryoSPARC and 3D refinement in RELION v5 with enabled Blush regularization, resulted in a final map at 3.2-Å resolution (Figs 1, 2A, S1 Table) displaying an impressive improvement in isotropy, as evidenced by the 10-fold increase in the cFAR score (Fig 2B), and overall interpretability (S3 Fig). Most importantly, post-processing in RELION yielded a sharpened map with clear main chain connectivity and signal for most side chains, allowing complete model building of a 2:1 $ALK_{TG-EGFL}$-ALKAL2 complex that includes the membrane-proximal EGFL-like domains of ALK. While the majority of the 2:1 $ALK_{TG-EGFL}$-ALKAL2 cryo-EM map has a local resolution to better than 3.5-Å resolution (Fig 1F), the C-terminal tips of the EGFL domains are only resolved at 4–4.5 Å resolution pointing to their inherent flexibility.

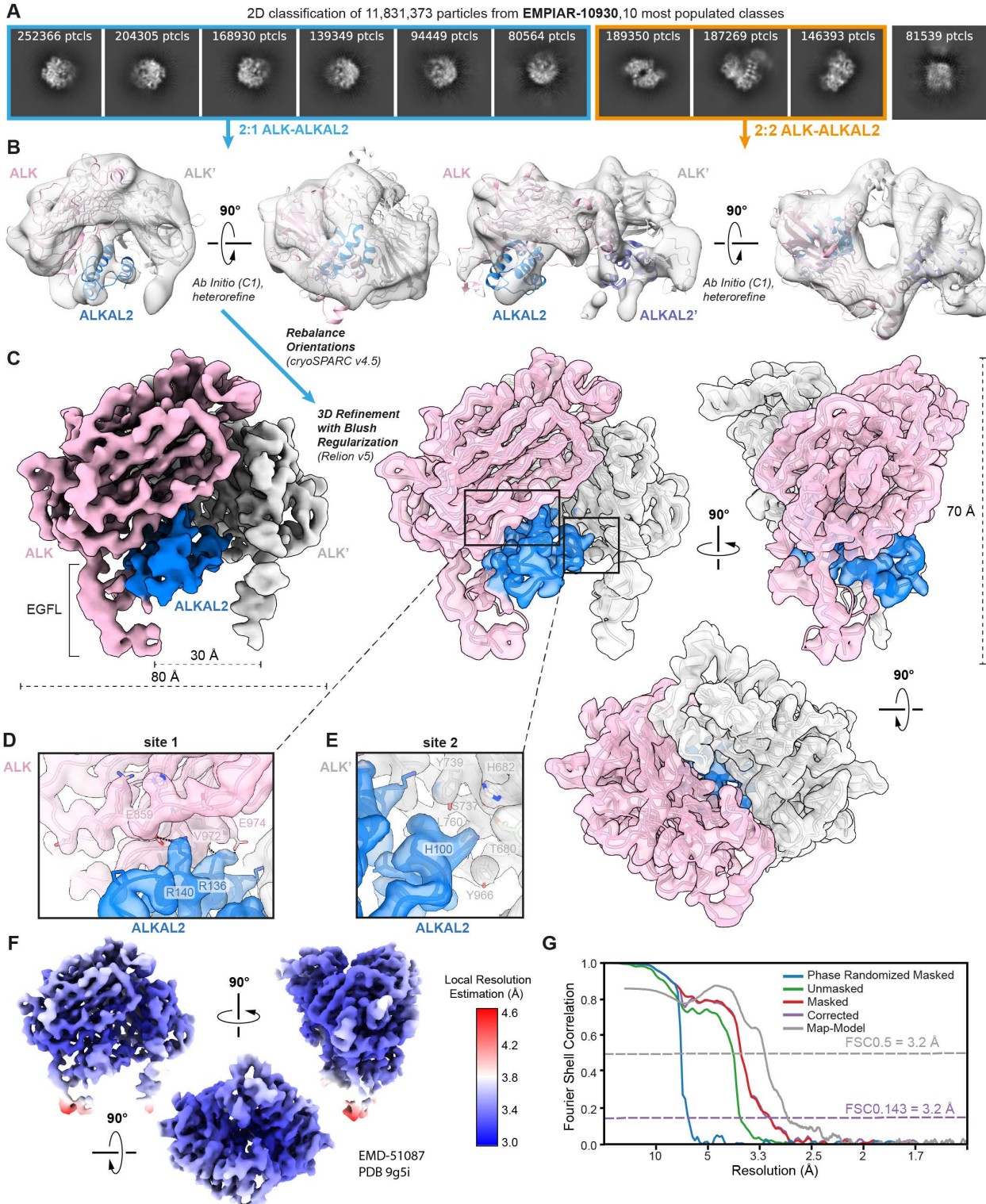

**Fig 1. Reanalysis of cryo-EM data from EMPIAR entry EMPIAR-10930 deposited by Reshetnyak and colleagues, Nature (2021) [7]. (A)** Ten most populated 2D class averages after performing 2D classification on 11.8 million selected particles. Classes corresponding to 2:1 ALK–ALKAL2 particles are highlighted in blue, while classes corresponding to 2:2 ALK–ALKAL2 particles are highlighted in orange. **(B)** Ab initio model generation

in cryoSPARC [18] results in classes corresponding to both a 2:1 ALK–ALKAL2 and 2:2 ALK–ALKAL2 assembly. Hetero-refinement of the 2:1 and 2:2 ALK–ALKAL2 classes results in the two models displaying 2:1 and 2:2 ALK–ALKAL2 assemblies, respectively. The ALK–ALKAL2 complexes with 2:1 (PDB 7nwz) and 2:2 (PDB 7n00) stoichiometries are shown fitted inside the maps, with ALK colored pink/gray and ALKAL2 colored blue/purple. **(C)** 3D refinement with Blush regularization [19] in RELION v5 of a trimmed particle set after extensive automatic rebalancing of particle orientations as implemented in cryoSPARC 4.5 (see also Materials and methods and S1 Table). A front view is shown of a map sharpened in RELION using a B-factor of −100 Å$^2$, with ALK colored pink/gray and ALKAL2 colored blue. Front, side, and top views are shown of a transparent map, sharpened using deep-EMhancer, with fitted structural model. **(D and E)** Zoomed-in insets illustrating the quality of the cryo-EM map corresponding to the two major interaction interfaces (site 1 and 2) in the ALK:ALKAL2 complex. Key interacting residues are annotated and shown as sticks. **(F)** Local resolution estimation performed in RELION v5 of the final map for the 2:1 ALK-ALKAL 2 complex. **(G)** Fourier shell correlation (FSC) plot corresponding to the final map of the 2:1 ALK:ALKAL2 complex. The corrected FSC (purple lines) is calculated after performing correction by noise substitution [20], and the resolution at FSC = 0.143 is annotated via a dotted purple line. A map-to model FSC curve calculated using the fitted model and an unsharpened, unfiltered full map is shown (gray line) with the resolution at FSC = 0.5 annotated via a dotted gray line.

To verify that more than half of the ALK-ALKAL2 particles present in the EMPIAR-10930 dataset indeed correspond to particles with a 2:1 stoichiometry, we pursued three independent heterogeneous 3D refinement runs in cryoSPARC using the full set of 11.8 million particles obtained after removing junk classes during the first 2D classification run (Materials and methods), and utilizing the final 2:1 and 2:2 ALK-ALKAL2 complex maps blurred to 20 Å as reference input volumes (S4 Fig). These three runs, each started using different generated random seeds, resulted in 53.5%, 53.1%, and 52.7% particles corresponding to the 2:1 ALK-ALKAL2 3D class, and 46.5%, 46.9%, and 47.3% to the 2:2 ALK-ALKAL2 3D class, thus corroborating the results observed during initial 2D classification (S4 Fig).

We next wondered whether 3D refinement with Blush regularization would be sufficient to obtain an isotropic map from the particle set without Orientation Rebalancing. Starting from the same particle set before the first Orientation Rebalancing step, followed by *Ab-Initio* model generation (3 classes), hetero-refinement and a final 3D refinement with Blush Regularization of the highest resolution class, resulted in a map at higher resolution (2.7 Å) but with lower isotropy (cFAR score of 0.1) than the map obtained by combining Orientation Rebalancing and 3D refinement with Blush regularization (Fig 3A and 3B). We note that 3D refinement with Blush regularization of the non-Orientation Rebalanced particle set (cFAR score of 0.1) is superior in improving map isotropy than NU-refinement without or with Orientation Rebalancing, which yields maps with cFAR scores of 0.01 and 0.08, respectively (Fig 3B). Furthermore, we also performed a 3D refinement in RELION starting from the Orientation Rebalanced particle set, but without enabling Blush regularization (Fig 3A and 3B). This refinement resulted in a more isotropic 3D reconstruction (cFAR score of 0.25) than NU-refinement in CryoSPARC after Orientation Rebalancing but stalled at a lower resolution of 4.3 Å, prohibiting de novo model building. Thus, enabling Blush regularization during 3D refinement of the Orientation Rebalanced particle set seems to both result in increased map isotropy as well as resolution. Taken together, our analysis demonstrates that the combination of extensive Orientation Rebalancing and 3D refinement with Blush regularization may serve as a suitable strategy to overcome severe preferred particle orientations *In Silico* in datasets that contain a full but extremely unbalanced range of orientations.

## Discussion

The new structure of ALK$_{TG-EGFL}$-ALKAL2 with 2:1 stoichiometry presented here is a critical contribution to the present collection of ALK/LTK-cytokine complexes revealed by X-ray crystallography and cryo-EM. The elucidation of an ALK$_{TG-EGFL}$-ALKAL2 complex with 2:1 stoichiometry based on a dataset that was initially reported to only support ALK$_{TG-EGFL}$-ALKAL2 complexes with 2:2 stoichiometry underscores the power of modern cryo-EM to harness the conformational and stoichiometric diversity harbored in protein samples. Furthermore, our analysis emphasizes the importance of public deposition of raw (cryo-EM) data to allow reanalysis and interpretation. Most importantly, though, this new structure is now the most complete ALK-ALKAL2 complex with 2:1 stoichiometry to date and agrees closely with the ALK$_{TG}$-ALKAL2 with 2:1 stoichiometry reported by De Munck *and colleagues* [5] featuring site 1 (ALK-ALKAL2), site 2 (ALK'-ALKAL2), and site 3 (ALK-ALK') interaction interfaces (Fig 1). Accordingly, the root-mean-square deviation (r.m.s.d.) between these

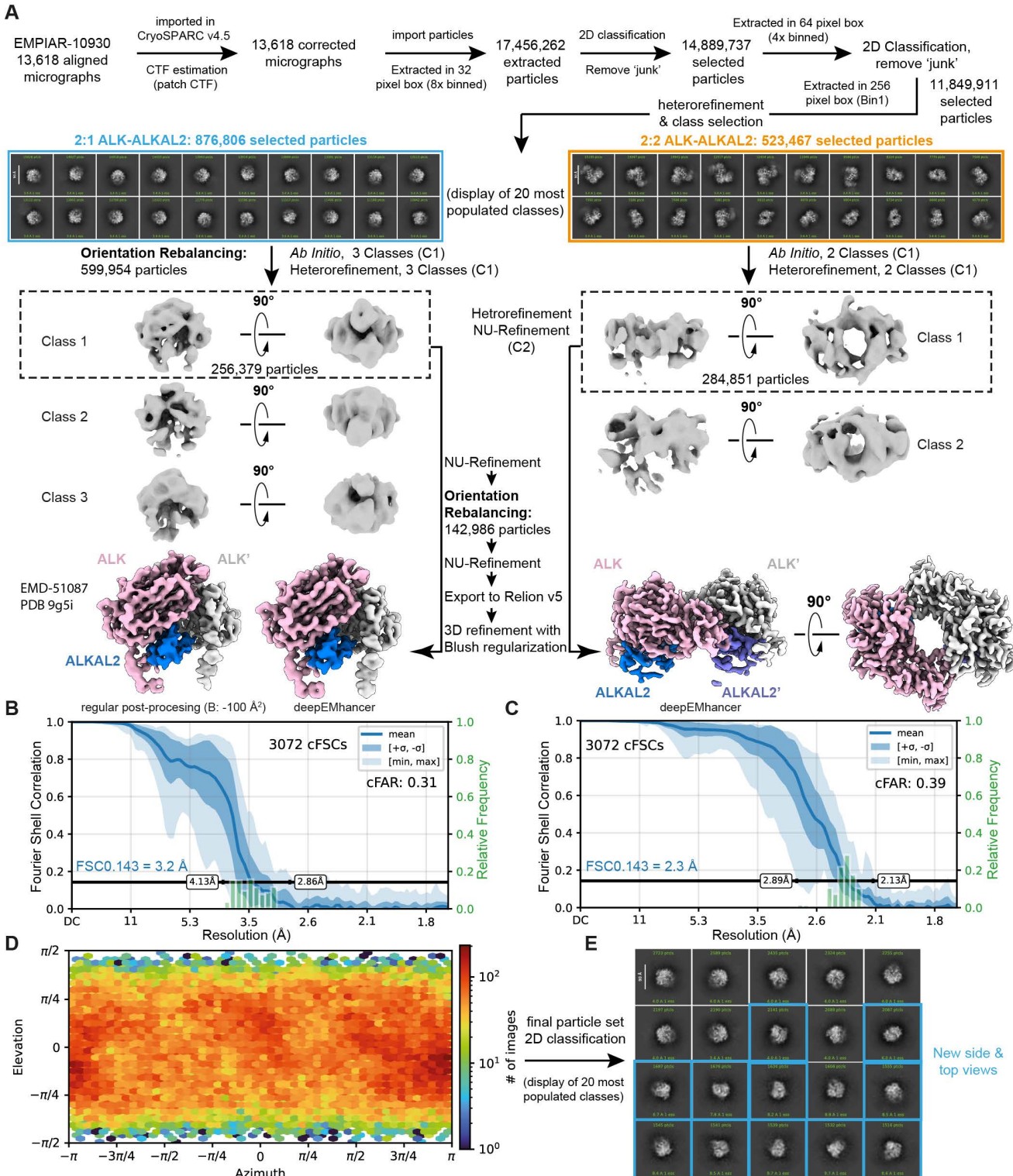

**A**

EMPIAR-10930 13,618 aligned micrographs → imported in CryoSPARC v4.5 / CTF estimation (patch CTF) → 13,618 corrected micrographs → import particles / Extracted in 32 pixel box (8x binned) → 17,456,262 extracted particles → 2D classification / Remove 'junk' → 14,889,737 selected particles → Extracted in 64 pixel box (4x binned) / 2D Classification, remove 'junk' → Extracted in 256 pixel box (Bin1) → 11,849,911 selected particles

heterorefinement & class selection

**2:1 ALK-ALKAL2: 876,806 selected particles**

(display of 20 most populated classes)

**2:2 ALK-ALKAL2: 523,467 selected particles**

Orientation Rebalancing: 599,954 particles

*Ab Initio*, 3 Classes (C1)
Heterorefinement, 3 Classes (C1)

*Ab Initio*, 2 Classes (C1)
Heterorefinement, 2 Classes (C1)

Class 1 — **90°** — 256,379 particles

Hetrorefinement NU-Refinement (C2)

Class 1 — **90°** — 284,851 particles

Class 2 — **90°**

Class 2 — **90°**

Class 3 — **90°**

NU-Refinement
**Orientation Rebalancing:** 142,986 particles
NU-Refinement
Export to Relion v5
3D refinement with Blush regularization

EMD-51087 PDB 9g5i

ALK  ALK'  ALKAL2

regular post-procesing (B: -100 Å²)  deepEMhancer

ALK  ALK'  ALKAL2  ALKAL2'  **90°**

deepEMhancer

**B**

Fourier Shell Correlation / Relative Frequency

3072 cFSCs
mean
[+σ, -σ]
[min, max]
cFAR: 0.31
FSC0.143 = 3.2 Å
4.13Å → 2.86Å
Resolution (Å): DC 11 5.3 3.5 2.6 2.1 1.8

**C**

3072 cFSCs
mean
[+σ, -σ]
[min, max]
cFAR: 0.39
FSC0.143 = 2.3 Å
2.89Å  2.13Å
Resolution (Å): DC 11 5.3 3.5 2.6 2.1 1.8

**D**

Elevation: π/2, π/4, 0, −π/4, −π/2
Azimuth: −π, −3π/4, −π/2, −π/4, 0, π/4, π/2, 3π/4, π
# of images: $10^0$ to $10^2$

**E**

final particle set 2D classification

(display of 20 most populated classes)

New side & top views

**Fig 2. Final processing pipeline of cryo-EM dataset EMPIAR-10930. (A)** Final processing steps were performed in CryoSPARC [18] v4.5 or RELION [22,23] v5. Post-processing was performed in RELION or using deepEMhancer [24]. **(B and C)** Conical FSC summary plots corresponding to the final maps of the 2:1 **(B)** and 2:2 **(C)** ALK:ALKAL2 complexes, generated via "Orientation Diagnostics" in cryoSPARC v4.5. The resolution at FSC = 0.143 is

annotated via dotted blue lines. **(D)** Azimuth plot showing the distribution of orientations over Azimuth (x-axis) and Elevation (y-axis) angles for the particle set corresponding to the final map of the 2:1 ALK-ALKAL2 complex, obtained after Orientation Rebalancing in cryoSPARC v4.5 and 3D refinement with Blush regularization [23] in RELION v5. **(E)** Display of the 20 most populated classes after 2D classification of the final particle stack corresponding to the 2:1 ALK-ALKAL2 complex. New side and top views not observed in initial 2D classification runs are shown as blue squares.

two structures is 1.3 Å over 675 aligned Cα-atoms (S6A Fig). The site 2 interface present in the 2:1 ALK:ALKAL2 complex, but not in the 2:2 ALK-ALKAL2 complex, involves the α1 helix of ALKAL2 with a key histidine residue (H100) interacting with S737, Y739, and L760 of ALK' (Fig 1E).

The membrane proximal EGFL domain of ALK engaged in the "site 1" interaction adopts a conformation identical to ALK found in the 2:2 ALK$_{TG-EGFL}$-ALKAL2 complex (r.m.s.d. = 0.554 Å over 345 aligned Cα-atoms, S6C Fig). Likewise, a 1:1 ALK$_{TG-EGFL}$-ALKAL2 binary complex extracted from the 2:2 ALK$_{TG-EGFL}$-ALKAL2 structure matches the "site 1" interface found in the 2:1 ALK$_{TG-EGFL}$-ALKAL2 cryo-EM model (r.m.s.d. = 0.631 Å over 400 aligned Cα-atoms, S6B Fig). Due to the asymmetric nature of the ALKAL2 monomer, the EGFL domain of the opposing ALK protomer (ALK') in the 2:1 ALK$_{TG-EGFL}$-ALKAL2 complex does not make any direct contacts at the "site 2" interaction with ALKAL2 (Fig 1). As a result, the ALK' EGFL domain is less resolved at its extremity due to increased flexibility, resulting in the absence of density for the minor β-hairpin in the cryo-EM map (Figs 1 and S6). Moreover, the major β-hairpin of ALK' has an outwards tilted conformation compared with the EGFL domain of ALK engaged in the "site 1" interaction (r.m.s.d.= 1.717 Å over 331 aligned Cα-atoms, S6D Fig). Despite this asymmetry, the C-termini of the EGFL domains of the two ALK copies in the 2:1 ALK$_{TG-EGFL}$-ALKAL2 complex obtained from EMPIAR-10930 are spaced ~30 Å apart (Fig 1), a distance that would allow facile juxtapositioning of receptor chains and TM helices as already illustrated for many RTK—cytokine assemblies featuring cytokine-mediated receptor dimerization such as KIT–SCF [10,11], CSF-1R–CSF-1 [12], Flt3–Flt3R [13], VEGF-VEGFR [16], PDGF-PDGFR [15], EGF–EGFR [14], FGF23–FGFR–αKlotho–HS [9], HER2–HER3–NRG1β [25], and Insulin bound to insulin receptor (IR) family receptors [26,27]. The presence of two distinct stoichiometries of the ALK-ALKAL2 complex among the particles imaged in the EMPIAR-10930 dataset is intriguing and their role in ALK signaling is currently unclear. One hypothesis could be that increasing local cytokine concentrations could shift an ALK-ALKAL2 assembly from a 2:1 to a 2:2 stoichiometry, resulting in an increase in the distance between the membrane proximal EGFL domains from 30 Å to 90 Å, possibly leading to different signaling outcomes (S5 Fig). Interestingly, designed ligands for erythropoietin receptor aimed at tuning the distance between receptor monomers upon dimerization result in modulation of erythropoietin receptor signaling and phosphorylation efficiency of downstream adaptors [28].

Full-length wild-type human ALKAL2 can form a dimer in solution via Cys66 in the predicted α-helix of its flexible N-terminal tail, yet monomeric ALKAL2 lacking this tail is as active as its dimeric counterpart [2]. While our study by De Munck *and colleagues* [5] and the study by Reshetnyak *and colleagues* [7] used monomeric or mutated (C66Y) ALKAL2, respectively, both 2:1 and 2:2 ALK-ALKAL2 assemblies present in EMPIAR-10930 are structurally compatible with dimeric ALKAL2. In the 2:2 ALK-ALKAL2 assembly, the additional 2 × 68 N-terminal ALKAL2 residues not built in the cryo-EM structure would need to be tacked underneath or along the length of the complex parallel to the cell membrane. On the other hand, in the 2:1 ALK-ALKAL2 assembly, the N-terminal tail would have ample space to flare outwards, allowing its facile linkage to another 2:1 ALK-ALKAL2 complex.

Finally, structural, biophysical, and cellular studies, including full-length ALK-ALKAL2 complexes, will be required to fully define the possible roles of distinct ALK/LTK-cytokine stoichiometries.

## Materials and methods

### Reanalysis of cryo-EM data as reported in EMPIAR-10930

Initial data analysis was carried out in CryoSPARC [18] v3.3.1, and additional steps were performed in either cryoSPARC [18] v4.5 or RELION [22,23] v5, specifically for the use of Orientation Rebalancing ("Rebalance Orientations" job) and

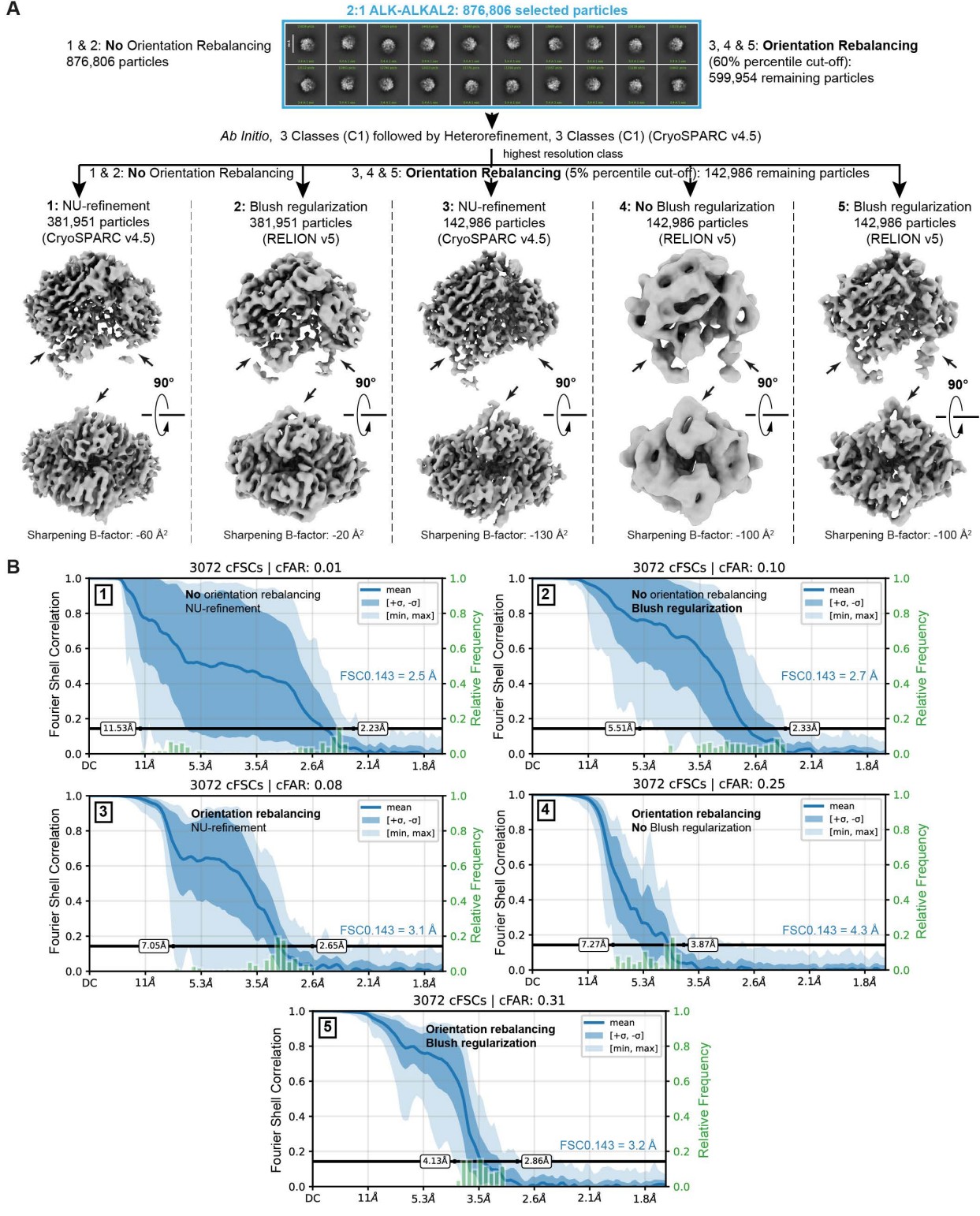

**Fig 3. Comparison of map (an)isotropy following data processing strategies with or without Orientation Rebalancing and/or 3D refinement with enabled Blush Regularization. (A)** Processing workflows (1–5) of cryo-EM dataset EMPIAR-10930 starting from 876,806 selected particles as outlined in Methods. Workflow 1 does not include Orientation Rebalancing (CryoSPARC v4.5) nor 3D refinement with Blush regularization (RELION

v5). Workflow 2 is the same as 1 but with a final 3D refinement with Blush regularization in RELION v5 instead of NU-refinement in CryoSPARC v4.5. Workflow 3 utilizes Orientation Rebalancing in CryoSPARC v4.5 followed by NU-refinement in CryoSPARC. Workflow 4 includes Orientation Rebalancing in CryoSPARC v4.5 followed by regular 3D refinement in RELION (without Blush regularization), while workflow 5 combines Orientation Rebalancing in CryoSPARC with 3D refinement with enabled Blush regularization in RELION. **(B)** Conical FSC summary plots for the final 3D refinements obtained in processing workflows 1, 2, 3, 4, and 5, generated via "Orientation Diagnostics" in cryoSPARC v4.5. The resolution at FSC = 0.143 is annotated in blue.

3D refinement including Blush regularization [19], respectively. Dose-weighted and motion-corrected micrographs were downloaded from https://www.ebi.ac.uk/empiar/EMPIAR-10930/ and were CTF-corrected using patch CTF estimation. After importing the uncleaned particle stack (~18 million particles) from EMPIAR-10930 and extracting the corresponding particles (8× binned), several iterative cycles of reference-free 2D classification were performed to get rid of "junk" particles, resulting in a "cleaned" set comprising 11,849,911 (~11.8 million) particles. This number of particles approaches the total number of particles (~12.4 million) used by Reshetnyak *and colleagues* [7]. Particle unbinning followed by reference-free 2D classification revealed that almost 2/3 of the 10 most populated classes do not correspond to classes shown in the final 2D classification represented in Reshetnyak *and colleagues* [7] (Fig 1A). Ab initio models generated on a selected subset of most populated classes (4× binned, 1.5 million particles, 3 Ab Initio classes, C1 symmetry) followed by a round of heterorefinement revealed one model that clearly corresponds to the 2:2 ALK–ALKAL2 structure reported by Reshetnyak *and colleagues*[7], as well as a model resembling the 2:1 ALK–ALKAL2 structure presented in De Munck *and colleagues* [5] (Figs 1B and S2). Next, the three Ab Initio classes were used together with the cleaned dataset of ~11.8 million particles (unbinned), as input for heterorefinement (C1, 9 classes). Classes 1 and 2 of this heterorefinement were grouped together, and selected particles (~1.2 million) were used for 3D classification without alignment (8 classes). Class 5 after 3D classification was selected (334,654 particles) and used as an input for a final Non-Uniform (NU) refinement [29], While the final map shows the presence of two ALK copies and one ALKAL2 copy (with its characteristic three central α-helices), the map suffers from the severe preferential particle orientations of the 2:1 ALK–ALKAL2 particles, as evidenced by the paucity of different views in the 2D classes as well as the Azimuth versus Elevation plot shown in S2 Fig. Indeed, Orientation Diagnostics, as implemented in the newest version of cryoSPARC [18] (v4.5), pointed to extreme anisotropy corresponding to a cFAR score of 0.03 (S2 Fig). Accordingly, the anisotropy present in the obtained NU-refined map of the 2:1 ALK$_{TG-EGFL}$–ALKAL2 complex prohibited further model building and refinement.

Next, starting from the heterorefinement of 11.849.911 particles, highest resolution 3D classes were grouped in two pools based on the presence or absence of 2-fold symmetry. Further processing of the particles with 2-fold symmetry (523,467 particles) included one additional round of 2D classification to remove remaining junk classes, *ab-Initio* model generation (2 classes), hetero-refinement and 3D NU-refinement of the highest resolution class while applying C2 symmetry, and resulted in a 2.3 Å resolution map for the 2:2 ALK$_{TG-EGFL}$-ALKAL2 complex (Fig 2). Further processing of the particles without apparent symmetry (876,806 particles) by 2D classification showed that 2D classes mainly consisted of side views of the 2:1 ALK-ALKAL2 complex, with a lack of discernible top views (Fig 2). We next used Orientation Rebalancing in cryoSPARC v4.5 to trim overpopulated views in the particle stack using a rebalance percentile threshold of 60, resulting in 599.954 remaining particles. These particles were used as an input for an Ab Initio refinement job (C1, 3 classes) followed by heterorefinement (C1, 3 classes). The highest resolution 3D class (class 1) was used for NU-refinement, followed by another round of Orientation Rebalancing, now using a more stringent rebalance percentile threshold of 5. Retained particles (142.986) where used for a second NU-refinement and subsequently exported to RELION v5 for a final round of 3D refinement with enabled Blush regularization. A final 2D classification was performed on the final particle stack (142.986 particles) to demonstrate the appearance of new side and top views not seen in previous 2D classification runs (Fig 2). Orientation Diagnostics jobs were run on obtained intermediate 3D reconstructions as well as the final map to monitor improvements in map isotropy. The final obtained map has a Fourier Shell Correlation (FSC, computed from independently refined half-maps) resolution of 3.2 Å at the 0.143 threshold (Fig 1, S1 Table) and was post-processed either in RELION using a sharpening B-factor of −100 Å$^2$ or by using deepEMhancer [24].

 

To obtain the ratio of particles in the dataset corresponding to 2:1 or 2:2 ALK-ALKAL2 complexes, a 3D heterorefinement run (CryoSPARC v4.5) was performed in triplicate, using the final maps for the 2:1 and 2:2 ALK-ALKAL2 complexes blurred to 20 Å and cleaned particle stack of 11,849,911 particles as input. Each run was initialized using a different random seed (run 1: 1740971936, run 2: 1817490782, run 3: 150017633).

## Model building and refinement

Two copies of ALK$_{TG-EGFL}$ and one copy of ALKAL2 were extracted from PDB ID 7n00, and rigid-body fitted in the sharpened map of the 2:1 ALK$_{TG-EGFL}$-ALKAL2 complex (B-factor: −100 Å$^2$) using USCF Chimera [30]. The resulting rigid-body fitted model was subjected to a cycle of manual building in Coot [31], aided by both the regularly sharpened map and a map sharpened using DeepEMhancer [24]. Next, the manually built and corrected model was real-space refined against the regularly sharpened map in Phenix v1.19.2-4158 [32], using global minimization, local grid search, atomic displacement parameter refinement, secondary structure restraints, and Ramachandran restraints. Several cycles of manual building in Coot were followed by real-space refinement in Phenix. After the last cycle of manual building in Coot, a nonbonded-weight parameter of 200 was used during refinement in Phenix. The resulting model was subjected to a final refinement against the two half-maps in REFMAC-Servalcat [33] run within CCP-EM suite v1.6 [34], with automatic weighting applied. The final model has a map-model FSC of 3.20 Å at the 0.5 threshold, calculated using the unfiltered, unsharpened full map (Fig 1). A summary of cryo-EM data collection, processing, refinement, and validation statistics can be found in S1 Table.

## Supporting information

**S1 Fig. Comparison of 2:1 ALK/LTK and 2:2 ALK receptor-ligand complexes. (A)** Cartoon views of structures of ALK-ALKAL2 and LTK-ALKAL1 complexes with 2:1 stoichiometry as determined by De Munck and colleagues [5] (PDB entries 7nwz and 7nx0) and **(B)** of ALK-ALKAL2 with 2:2 stoichiometry as determined by Reshetnyak and colleagues [7] and Li and colleagues [6] (PDB entries 7n00 and 7ls0). Putative positions of the EGFL domains absent in **(A)** are shown as dotted outlines.
(TIF)

**S2 Fig. Initial processing workflow of cryo-EM dataset EMPIAR-10930. (A)** All initial processing steps were performed in CryoSPARC [18] v3.3.1. **(B)** Fourier Shell Correlation (FSC) plot corresponding to the final map obtained after non-uniform refinement (C1 symmetry). Curves are shown after applying no mask (blue), a loose mask (green), or a tight mask (red) to the two half-maps before calculating FSCs. The corrected FSC (purple) is calculated using the tight mask with correction by noise substitution [20], and the resolution at FSC = 0.143 is annotated via a dotted purple line. **(C)** Conical FSC summary plot generated via "Orientation Diagnostics" in cryoSPARC v4.5. **(D)** Left: Azimuth plot showing the distribution of orientations over Azimuth (x-axis) and Elevation (y-axis) angles for the particle set corresponding to the NU-refined map shown in **(A)**. Right: Plot showing Relative Signal amount vs. Viewing Direction.
(TIF)

**S3 Fig. Representative EM map regions for site 1, site 2, and site 3 interfaces of the 2:1 ALK-ALKAL2 complex. (A)** Front (upper view) and top (lower view) views are shown of a transparent map of the 2:1 ALK-ALKAL2 complex with fitted structural model, with ALK colored pink/gray and ALKAL2 colored blue. Insets shown in **(B)**, **(C)**, and **(D)** are annotated with black squares. **(B and C)** Insets showing zooms of the site 1 ALK-ALKAL2 **(B)**, site 2 ALK'-ALKAL2 **(C)**, and site 3 ALK-ALK' **(D)** interfaces, with the EM map displayed as a dark blue mesh. The fitted 2:1 ALK-ALKAL2 model is shown as a cartoon with selected annotated residues shown as sticks (ALK colored pink/gray and ALKAL2 colored blue).
(TIF)

**S4 Fig.  3D heterorefinement using a set of 11,849,911 particles and the final 2:1 and 2:2 ALK-ALKAL2 maps blurred to 20 Å as input.** All initial processing steps were performed in CryoSPARC [18] v3.3.1. The three 3D heterore-finement replicate runs, each ran with a different random seed, were performed using CryoSPARC v4.5.
(TIF)

**S5 Fig.  Side-by-side comparison of 2:1 and 2:2 ALK-ALKAL2 complexes present in the EMPIAR-10930 cryo-EM dataset.** Models with pdb code 9g5i (this study) and 7n00 (Reshetnyak et al., 2021) [7] are shown as cartoons with ALK/ALK' colored pink/gray and ALKAL2 colored blue in the 2:1 ALK-ALKAL2 complex, or blue/purple in the 2:2 ALK-ALKAL2 complex. The distance between the C-termini of membrane-proximal EGFL domains is annotated.
(TIF)

**S6 Fig.  Structural alignments of ALK-ALKAL2 X-ray and cryo-EM models. (A)** Structural alignment of the 2:1 ALK-ALKAL2 X-ray structure (PDB 7nwz) with the cryo-EM structure of the 2:1 ALK-ALKAL2 complex obtained in this study (PDB 9G5I). **(B)** Structural alignment of a 1:1 ALK-ALKAL2 complex, extracted from the cryo-EM structure of the 2:2 ALK-ALKAL2 complex (PDB 7n00), with the cryo-EM structure of the 2:1 ALK-ALKAL2 complex obtained in this study. **(C)** Structural alignment of one copy of ALK, extracted from the cryo-EM structure of the 2:2 ALK-ALKAL2 complex (PDB 7n00), with the first ALK molecule (pink) present in the cryo-EM structure of the 2:1 ALK-ALKAL2 complex obtained in this study. **(D)** Structural alignment of one copy of ALK, extracted from the cryo-EM structure of the 2:2 ALK-ALKAL2 complex (PDB 7n00), with the second ALK molecule (ALK', gray) present in the cryo-EM structure of the 2:1 ALK-ALKAL2 complex obtained in this study.
(TIF)

**S1 Table.  Cryo-EM data collection, processing, refinement, and validation statistics.**
(DOCX)

**S1 File.  Particle coordinates .star file, obtained after final 3D refinement with Blush Regularization in RELION 5.**
(ZIP)

## Author contributions

**Conceptualization:** Jan Felix, Savvas N. Savvides.

**Formal analysis:** Jan Felix, Steven De Munck, J. Fernando Bazan, Savvas N. Savvides.

**Funding acquisition:** Savvas N. Savvides.

**Investigation:** Jan Felix.

**Supervision:** Savvas N. Savvides.

**Validation:** Jan Felix.

**Visualization:** Jan Felix, Steven De Munck.

**Writing – original draft:** Jan Felix, Savvas N. Savvides.

**Writing – review & editing:** Jan Felix, Steven De Munck, J. Fernando Bazan, Savvas N. Savvides.

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
