## [Editor Report · Decision Letter 0]

16 Jan 2025

Dear Dr Savvides,

Thank you for submitting your manuscript entitled "Overcoming cryo-EM map anisotropy reveals ALK-cytokine assemblies with distinct stoichiometries" for consideration as a Short Report by PLOS Biology.

Your manuscript has now been evaluated by the PLOS Biology editorial staff, as well as by an academic editor with relevant expertise, and I am writing to let you know that we would like to send your submission out for external peer review.

Once your full submission is complete, your paper will undergo a series of checks in preparation for peer review. After your manuscript has passed the checks it will be sent out for review. To provide the metadata for your submission, please Login to Editorial Manager (https://www.editorialmanager.com/pbiology) within two working days, i.e. by Jan 18 2025 11:59PM.

Kind regards,

Richard

Richard Hodge, PhD

rhodge@plos.org

PLOS

---

## [Decision Letter · Decision Letter 1]

20 Feb 2025

Dear Dr Savvides,

Thank you for your patience while your manuscript "Overcoming cryo-EM map anisotropy reveals ALK-cytokine assemblies with distinct stoichiometries" went through peer-review at PLOS Biology. I'm handling your paper temporarily while my colleague Dr Richard Hodge is out of the office. Your manuscript has now been evaluated by the PLOS Biology editors, an Academic Editor with relevant expertise, and by four independent reviewers.

In light of the reviews, which you will find at the end of this email, we are pleased to offer you the opportunity to address the comments from the reviewers in a revision that we anticipate should not take you very long. We will then assess your revised manuscript and your response to the reviewers' comments with our Academic Editor aiming to avoid further rounds of peer-review, although we might need to consult with the reviewers, depending on the nature of the revisions.

In light of the reviews, which you will find at the end of this email, we are pleased to offer you the opportunity to address the comments from the reviewers in a revision that we anticipate should not take you very long. We will then assess your revised manuscript and your response to the reviewers' comments with our Academic Editor aiming to avoid further rounds of peer-review, although we might need to consult with the reviewers, depending on the nature of the revisions.

IMPORTANT: In addition, I would be grateful if you could please address the following editorial and data-related requests that I have provided below (A-F):

(A) We routinely suggest changes to titles to ensure maximum accessibility for a broad, non-specialist readership. In this case, we would suggest a minor edit to the title, as follows. Please ensure you change both the manuscript file and the online submission system, as they need to match for final acceptance: “Reanalysis of cryo-EM data overcoming map anisotropy reveals ALK-cytokine assemblies with both 2:1 and 2:2 stoichiometries” (assuming that this is still accurate)

(B) Thank you for providing the structural data in the PDB and EMDB database (9G5I and EMD51087). However, we note that the structural data is currently on hold for release. We ask that you please make the structures publicly available at this stage before publication.

(C) Please also ensure that each of the relevant figure legends in your manuscript include information on *WHERE THE UNDERLYING DATA CAN BE FOUND*, and ensure your supplemental data file/s has a legend.

(D) Per journal policy, if you have generated any custom code during the course of this investigation, please make it available without restrictions. Please ensure that the code is sufficiently well documented and reusable, and that your Data Statement in the Editorial Manager submission system accurately describes where your code can be found. NOTE: we cannot accept sole deposition of code in GitHub, as this could be changed after publication. However, you can archive this version of your publicly available GitHub code to Zenodo. Once you do this, it will generate a DOI number, which you will need to provide in the Data Accessibility Statement (you are welcome to also provide the GitHub access information). See the process for doing this here: https://docs.github.com/en/repositories/archiving-a-github-repository/referencing-and-citing-content

(E) Please note that per journal policy, the model system/species studied should be clearly stated in the abstract of your manuscript.

(F) Please ensure that your Data Statement in the submission system accurately describes where your data can be found and is in final format, as it will be published as written there.

**IMPORTANT - SUBMITTING YOUR REVISION**

*Resubmission Checklist*

*Published Peer Review*

*PLOS Data Policy*

*Blot and Gel Data Policy*

Sincerely,

Roli Roberts

Roland G Roberts PhD

Senior Editor

PLOS Biology

rroberts@plos.org

on behalf of

Richard Hodge, PhD

Senior Editor

PLOS Biology

rhodge@plos.org

REVIEWER'S COMMENTS:

Reviewer #1:

Felix et al. reprocessed the cryo-EM dataset of the ALK-ALKAL2 complex, which was deposited at EMPIAR. By employing blush regularization, a new feature in RELION5, the authors successfully resolved the structure of the 2:1 ALK-ALKAL2 complex at an overall resolution of 3.2 Å. Previously, this structure could not be determined due to the particles' strong preferred orientation. The resolved 2:1 ALK-ALKAL2 complex is virtually identical to the previously determined crystal structure, supporting the notion that this stoichiometry represents a physiologically relevant state. Therefore, this study provides valuable insights into the activation mechanism of the ALK receptor. The quality of the reported cryo-EM structure is high, and the model appears to be reliable. I recommend its publication as a short report in PLOS Biology.

I have one suggestion for the authors to consider: The paper mentions a hypothesis that increasing local cytokine concentrations might shift an ALK-ALKAL2 assembly from a 2:1 to a 2:2 stoichiometry, increasing the distance between the membrane-proximal domains from 30 Å to 90 Å, potentially leading to different signaling outcomes. This intriguing idea can be easily tested. Can the authors examine ALK activation in response to varying concentrations of ALKAL2? If the hypothesis is correct, a bell-shaped dose-response curve might be observed.

Reviewer #2:

[identifies himself as Moosa Mohammadi]

In this manuscript, Felix and colleagues seek to address the ongoing debate regarding mechanism of ligand-driven dimerization/activation of ALK receptor tyrosine kinase family members, ALK and LTK. Previously, this group reported X-ray crystal structures of ALK and LTK ectodomains complexed with their cognate ligands, ALKAL2, and ALKAL1, respectively, revealing how a single ligand induces asymmetric receptor dimerization with 1:2 ligand-receptor stoichiometry reminiscent of the growth hormone paradigm. In contrast, based on the cryoEM analysis of the ALKAL2-ALK complex, Reshetnyak and colleagues proposed a symmetric 2:2 model of dimerization for ALK activation. Similarly, Li et al., reported the X-ray structure of a 2:2 ALKAL2-ALK symmetric dimer using an ALK-ALKAL2 fusion protein. To understand the discord between these structures, Felix and colleagues reanalyzed the raw cryo-EM data deposited by Reshetnyak et al. (access code EMPIAR-56 10930) and found that the bulk (~60%) of ALKAL2-ALK particles arrange as 1:2 complexes as seen in 2D classification. However, these particles suffer from severe preferential particle orientations leading to highly anisotropic maps which preclude model building and detailed structural interpretations. The authors mitigated this problem by performing exhaustive automatic particle orientation rebalancing as implemented in the latest version of cryoSPARC to trim over-populated views, followed by 3D refinement with Blush regularization within Relion. As a result, the authors were able to achieve a dramatically improved map at 3.2 Å resolution allowing them to confidently build a 1:2 ALKAL2-ALK model resembling that seen in their previous X-ray structures.

This study has been conducted meticulously and clearly demonstrates that the asymmetric mode of receptor dimerization is a common mechanism by which ALK/LTK signaling is initiated at the cell surface. The 2:2 symmetric dimer seen in EMPIAR-56 10930 dataset is likely due to the presence of saturating ligand concentration in the sample preparation.

Comments:

1) Page 4, line 139: A comparable distance (~30-40 Å) is seen between the membrane insertion points of the two extracellular domains of FGF receptor chains in the recently published cryoEM structures of three distinct 1:2:1:1 FGF23-FGFR-�Klotho-HS quaternary assemblies (Chen et al., Nature 2023). Notably, these structures feature asymmetric mode of receptor dimerization by the ligand. This should be cited.

2) Page 4, line 144: the typo "tot" should be corrected to "to".

3) The authors should briefly mention the adoption of symmetric and asymmetric modes of receptor dimerization by different RTK families, i.e., asymmetric (ALK, FGFR) and symmetric (EGFR, Kit, PDGFR, VEGFR).

Reviewer #3:

[identifies himself as Zhe Zhang]

Recent structural studies have shown different stoichiometries for ALK/LTK-cytokine complexes. Specifically, the authors' lab previously documented a 2:1 stoichiometry for ALK-ALKAL2 and LTK-ALKAL1 complexes, while two other groups reported a 2:2 stoichiometry for ALK-ALKAL2 complex. This discrepancy prompted the authors to reanalyze the cryo-EM data deposited in EMPIAR-10930, initially reported by Reshetnyak et al. to generate the 2:2 ALK-ALKAL2 complex structure, in order to resolve these differences.

The authors discovered that the dataset contained two distinct populations of particles corresponding to the 2:1 and 2:2 ALK-ALKAL2 complexes. Notably, the 2:1 complex exhibited severe preferred particle orientations, resulting in anisotropic maps. To address this issue, the authors utilized orientation rebalancing in cryoSPARC to trim overpopulated views, followed by 3D refinement using Blush regularization in RELION. These methods significantly improved the isotropy of the map, yielding a final 3.2 Å resolution map for the 2:1 ALK-ALKAL2 complex. This new cryo-EM structure provides clear evidence for the presence of the 2:1 stoichiometry alongside the previously reported 2:2 stoichiometry in the EMPIAR-10930 dataset. The existence of both stoichiometries suggests that ALK signaling may involve dynamic changes in receptor-cytokine interactions, potentially influenced by local cytokine concentrations.

Overall, this study addresses the previously conflicting structural data and highlights the importance of public data deposition for reanalysis. I am pleased to support its publication in PLOS Biology. I have a few suggestions for the authors to consider, which could further enhance the rigor of this work.

1. The authors identified the 2:1 and 2:2 ALK-ALKAL2 complexes solely based on the 2D classification results and concluded that these two populations represent 60% and 40% of the total particles, respectively. However, the reliability of this ratio is questionable. Firstly, it is uncertain whether the two populations can be confidently distinguished based on the blurry 2D images. Secondly, the repeatability of this result is also in doubt. The authors could consider performing a heterogeneous refinement in cryoSPARC using all the good 2D classes as input and taking the two high-resolution 3D models (2:1 and 2:2) as references. In this way, all the particles could be assigned to these two classes according to their similarity to these two structures. If the ratio remains consistent after several repeats, it would further support the authors' conclusion and provide a better evaluation of the ratio between the 2:1 and 2:2 complexes in this dataset.

2. As noted in the Discussion section, the RMSD between the new 2:1 ALK-ALKAL2 complex described in this study and the previously reported 2:1 crystal structure (7nwz) is 1.3 Å. Please include a superimposition of these two structures in the supplementary figure to substantiate this claim. If there are significant differences between these structures, it would be advisable to highlight and briefly discuss them.

3. Please include EM densities for some representative regions of the deposited structure in the supplementary figure.

4. In line 114, the cFAR score should be 0.08, as shown in Fig. 3B, rather than 0.8.

5. In Supplementary Table 1, the number of micrographs should be 13,618, not 13.618.

Reviewer #4:

[identifies himself as Arjen J. Jakobi]

This study investigates the structural mechanisms of ALK (Anaplastic Lymphoma Kinase) and LTK (Leukocyte Tyrosine Kinase) activation by their cytokines, ALKAL2 and ALKAL1, which have critical roles in development, metabolism, and cancer. Different stoichiometries of ALK/LTK-cytokine assemblies (2:1 and 2:2) have been observed via X-ray crystallography and cryo-EM; one set of original crystallography data were reported by the authors of this study. A reanalysis of the published cryo-EM data from another research team (EMPIAR-10930) by the authors revealed that 2:1 and 2:2 complexes co-occur in the published dataset and that many (in fact, a majority of) particles correspond to the 2:1 ALK-ALKAL2 complex, though these suffer from orientation bias and associated resolution anisotropy. Using recent advances in cryo-EM data processing, including a data-driven regularised refinement technique, the authors reconstructed the 2:1 ALK-ALKAL2 complex at ~3.2 Å overall resolution from the publicly deposited cryo-EM dataset. This work reconciles the structural understanding of ALK and LTK receptor dimerisation and, while acknowledging both 2:1 and 2:2 complexes do exist, proposes a unifying signaling-ready 2:1 model across both systems.

The (re)analysis and the study are thorough and solid, carefully performed and presented, and the manuscript is well written and illustrated. I enjoyed reading it. The authors present a series of structurally convincing arguments why the 2:1 complex appears to be a plausible arrangement also in a physiological context. The study not only adds a new (=more complete) 2:1 structure to the collection of experimentally determined structures of the ALK-ALKA2 complex, but also provides a beautiful example why public deposition of raw electron micrographs is so important for the community to allow re-analysis of previous data while computational tools develop and improve. I am limiting my comments to technical aspects regarding the analysis of cryo-EM data and data processing workflows as I lack expertise related to cytokine biology that would allow me to evaluate the biological conclusions drawn. Below, I summarise a few comments and suggestions that the authors may consider.

Minor comments:

1. A host of data and experiments suggest that unbalanced particle orientations in cryo-EM single particle samples can - more often than not - be attributed to preferential adsorption of the molecules at the air-water interface (AWI) of holey support films. Whilst the authors' analysis in the present study is perfectly solid and the evidence for a 2:1 complex in the EMPIAR-10930 data is unambiguous, the possibility remains that the observed 2:1 complex, which displays strong orientation bias, could also be a result of 2:2 complexes partly dissociated or otherwise structurally biased at the AWI. I acknowledge that this would involve a substantial conformational change to accommodate the relative orientation of both ALK monomers in the observed 2:1 ALK-ALKAL2 complex vs the 2:2 complex, which may not be obvious (and possibly unlikely) but as far as I infer from the accumulated data can also not be decisively refuted a priori. Another slightly worrying observation is that out of almost 12,000,000 cleaned-up particles after several rounds of 2D classification only about 10% (~870,000 for 2:1 + ~520,000 for 2:2) remain after 3D (hetero)refinement (cf. Figure 2A), which might suggest that a majority of particles is either structurally compromised or otherwise limited in contributing useful signal. Are all the discarded particles associated with "junk" classes? If so, what may this tell about the sample as a whole? If not, how were the final classes selected? None of the above observations question or invalidate the authors' conclusions in any way, but I invite the authors to consider and discuss these possibilities; or discuss why they can be discarded.

2. My previous question is also spurred by the authors' earlier observation of the 2:1 ALK-ALKAL2 complex in the crystal (PDB 7NWZ) but not in solution with SEC-MALS experiments in their original report (Munck et al., 2021); instead, a 1:1 complex was the abundant species in solution, in agreement with ITC experiments that also were consistent with 1:1/2:2 binding. In contrast, as observed in with the crystal structure, the cryo-EM data appears to suggest that the 2:1 stoichiometry is the dominant complex under these conditions. Are these, on first glance conflicting, observations unique to ALK-ALKAL2 or have similar intricacies also been observed in other cytokine/cytokine receptor systems? If the authors consider the cryo-EM experiments as representative for the behavior in bulk solution, how do can the the biochemical data be congruently rationalised with the observations from crystallography/cryo-EM? This is intriguing and some discussion to place these observations in context would be helpful to readers with limited knowledge of the cytokine field like me.

3. While the improvement of RELION refinement is evident, I feel it would be good to separate out the effect of Blush regularisation from regular RELION refinement to determine if indeed it is the additional regularisation that is beneficial/necessary in this case, since from Figure 3B it appears a major factor was orientation re-balancing. Currently the authors compare NU refinement in Cryosparc to Blush regularisation in RELION for particle sets before and after orientation re-balancing. The beneficial effect of RELION/Blush is most striking for the case before orientation re-balancing. I agree with the authors that comparing Cryosparc NU refinement to RELION+Blush is the most meaningful comparison as both strategies employ fundamentally similar, though conceptually different, ways of refinement regularisation. To separate out the effect of Blush regularisation over regular RELION refinement on improving isotropy of the reconstructed map, it should ideally be compared to a RELION 3D refinement without Blush regularisation using the same orientation-rebalanced particle set. Such an analysis would strengthen the assertion that the observed effect is indeed due to improved regularisation through Blush during refinement as opposed to a more generic difference in CryoSparc NU vs. regular RELION refinement.

4. Page 3: "…of similar resolution and informational content": I believe it should be "information content".

5. Page 3: "…a massively improved final map…": it is ambiguous what "massively improved" means or refers to. I suggest either dropping the adjective or making explicit what property/properties of the map has/have been improved (isotropy, interpretability, …).

6. Caption Figures 1,2; Supplementary Figures 2: "Gold-standard Fourier Shell Correlation (FSC) plots corresponding to the final maps…". Whilst broadly used, the term "gold-standard FSC" is not without controversy in the cryo-EM field. A neutral way could be simply stating: "Fourier Shell Correlation (FSC) plots computed from independently refined half maps of…".

---

## [Editor Report · Decision Letter 2]

23 Mar 2025

Dear Savvas,

On behalf of my colleagues and the Academic Editor, Yan Zhang, I am pleased to say that we can accept your manuscript for publication, provided you address any remaining formatting and reporting issues. These will be detailed in an email you should receive within 2-3 business days from our colleagues in the journal operations team; no action is required from you until then. Please note that we will not be able to formally accept your manuscript and schedule it for publication until you have completed any requested changes.

PRESS

Best wishes, 

Richard

Richard Hodge, PhD

rhodge@plos.org

PLOS
